# Fabricating Air Pressure Sensors in Hollow-Core Fiber Using Femtosecond Laser Pulse

**DOI:** 10.3390/mi14010101

**Published:** 2022-12-30

**Authors:** Changning Liu, Wuqiang Tao, Cong Chen, Yang Liao

**Affiliations:** 1College of Physics and Electronic Science, Hubei Normal University, Huangshi 435002, China; 2State Key Laboratory of High Field Laser Physics and CAS Center for Excellence in Ultra-Intense Laser Science, Shanghai Institute of Optics and Fine Mechanics (SIOM), Chinese Academy of Sciences (CAS), Shanghai 201800, China; 3Shanghai Key Laboratory of Modern Optical System, Engineering Research Center of Optical Instrument and System, Ministry of Education, School of Optical-Electrical and Computer Engineering, University of Shanghai for Science and Technology, Shanghai 200093, China

**Keywords:** femtosecond laser, sensor, anti-resonance, optical fiber

## Abstract

In this paper, a hollow core fiber was spliced with standard single-mode fibers to form a fiber optic gas pressure sensor, and its sensing characteristics with single hole or multi-holes punched on the hollow core fiber with femtosecond laser pulses were investigated. The experiments demonstrate that the air pressure sensitivity of the single hole sensor was −3.548 nm/MPa, with a linearity of 99.45%, while its response times for air pressure’s rise and fall were 4.25 s and 2.52 s, respectively. The air pressure sensitivity of the ten-hole sensor was up to −3.786 nm/MPa, with a linearity of 99.47%, while its response times for air pressure’s rise and fall were 2.17 s and 1.30 s, respectively. Theoretical analysis and experimental results indicate that the pressure sensitivity of the sensor with an anti-resonant reflecting guidance mechanism mainly comes from the refractive index change of the air inside the hollow core fiber. The proposed device with multi-holes drilled by a femtosecond laser has the advantages of fabrication simplicity, low cost, fast response time, good structural robustness, high repeatability, high sensitivity to air pressure, and insensitivity to temperature (only 10.3 pm/°C), which makes it attractive for high pressure sensing applications in harsh environments.

## 1. Introduction

Air pressure has been a key parameter in industrial production and meteorological monitoring. Recently, fiber-optic sensors for air pressure have been widely studied because of their advantages of small size, high sensitivity, and fast response time. Various solutions of fiber-optic air sensors have been proposed, including the gas pressure sensor based on hollow-core photonic bandgap fiber [1,2,3,4,5], inner air-cavity [6,7,8,9], a combination of hollow-core fiber and photonic crystal fiber [10,11], dual side-hole fiber [12], twin-core fiber [13], double off-axis twisted deformation of single mode fiber [14], quartz capillary tube with an open cavity [15], hollow optical fiber with two cores [16], laser-heated silicon pillar attached to a fiber tip [17], fiber-tip polyvinyl chloride diaphragm [18], and multimode fiber filled with ultraviolet adhesive [19]. However, these sensors usually need complex manufacturing processes and expensive specialty fibers. The hollow-core fiber (HCF) is not only cheap, simple to fabricate, but also of better compatibility with standard single-mode fibers (SMFs), making it suitable for fiber optic sensing applications [20,21,22,23,24,25,26,27,28,29].

In this paper, we propose and demonstrate a gas pressure sensor based on an anti-resonant reflecting guidance mechanism with HCF perforated by a femtosecond laser. The proposed sensor is fabricated by splicing a section of HCF between two sections of standard SMFs, and the single hole or multi-holes punched on the PCF enable the gas to enter or leave the cavity. The measurement of response time shows that the air pressure response of the fiber sensor with ten holes is twice as fast as that of the sensor with only one hole. Moreover, the proposed air pressure sensors also appear high sensitivity and a relatively low temperature cross-sensitivity of 2.9 kPa/°C and could be of a high potential in monitoring the gas pressure in high-temperature and high-pressure environments.

## 2. Fabrication of Sensors

Figure 1 shows a splicing process between a HCF with an inner diameter of 30 μm and a standard SMF (YOFC, 9 μm/125 μm). First, the coating layer of SMF is removed with a fiber stripper to expose its cladding layer, and the fiber is cleaved and cleaned to obtain a flat end face. The HCF is burned with a flame to remove its coating layer, and then is cleaved by a fiber cutting knife to obtain flat end faces as well. Then, both ends of the HCF are fused to the SMFs, resulting in a SMF-HCF-SMF structure, as illustrated in Step 3. As shown in Step 4, the air hole structure was fabricated by the femtosecond laser drilling in the hollow-core fiber fixed on a three-dimensional machining platform.

Figure 2a shows a microscopy image of a junction between a HCF with an inner diameter of 30 μm and a standard SMF. Figure 2b presents the interference spectra of the optical fiber sensor corresponding to a HCF with different lengths. It can be seen that the resonance peak intensity of the transmission spectrum gradually increases as the HCF length increases.

As shown in Figure 2b, the resonance peaks associated with the HCFs with the lengths of 1.4 mm and 2.5 mm are less than 5 dB, which are too low for accurate data analysis. When the length of the HCF extends to 10 mm, such fiber sensors are too big and not suitable for compact packaging. Fortunately, the resonance peak intensity of the HCF between 5 and 7 mm is greater than 10 dB, which fully meets the experimental requirements. In the subsequent experiments, the HCF with the length of about 6 mm was selected for fusion splicing with the SMF. The spliced structure was then subjected to laser drilling by a femtosecond laser (Coherent, Santa Clara, CA, USA) with the wavelength of 800 nm, the pulse width of 35 fs, and the laser energy of 50 μJ. The femtosecond laser pulses were focused on the middle top of the HCF with a 50× microscope objective, and a hole was ablated in less than 5 s with a programmed three-dimensional stage moving upward in the vertical direction at a speed of 10 μm/s. In addition, a refractive index matching liquid was applied to the HCF during both the femtosecond laser processing and microscopic imaging, which eliminated the adverse effects of spherical aberration on the experiments. The dimension (diameter and distance) accuracy of the small holes could be as small as 1 μm by use of a self-built micromachining station.

Figure 3 presents microscopy images of a single micro-hole in an HCF drilled by a femtosecond laser, showing the micro-hole has a round shape and a diameter of about 20 μm. It can be seen from Figure 3b that the micro-hole has almost vertical inner walls and the interior of the HCF is well connected to the outside atmosphere. Figure 4 presents microscopy images of ten micro-holes in an HCF drilled by a femtosecond laser. The ten micro-holes are equally arranged along the axial direction of the HCF, with a spacing of 100 μm between the holes and a diameter of 20 μm. All these images were taken by an Olympus BX 51 microscope under transmitted illumination. In order to verify the connectivity of the ten micro-holes, the HCF was filled with water to obtain a clear profile of the micro-holes, as shown in Figure 4b. Note that the dark region under the left micro-hole in Figure 4b is an interface between the air and water in the hollow core.

## 3. Theoretical Analysis

In this work, the sandwiched HCF between two SMFs can be considered as a single-layer anti-resonant reflecting optical waveguide (ARROW) [20,23]. The high refractive index cladding of HCF is regarded as a Fabry-Pérot etalon. When the wavelength of incident light is close to the resonance wavelength, light passes out of the high refractive index cladding, resulting in the periodic lossy dips in the transmission spectrum of HCF, as shown in Figure 2b. The propagating light at the anti-resonant wavelength (far away from the resonant wavelength) is internally reflected and confined in the hollow core of HCF, corresponding to the location of the periodic transmission peaks in Figure 2b.

The resonant wavelength can be simplified as the following expression [23]:(1)$$\lambda_{m}=\frac{2d}{m}\sqrt{n_{cl}^{2}-n_{air}^{2}},$$
where *m* is the resonant order, *d* is thickness of the cladding (~47.5 µm), *n_cl_* (~1.45), and *n_air_* (~1) are refractive indexes of the cladding and hollow core of the HCF, respectively.

According to Equation (1), we can derive the free spectral range (FSR):(2)$$\mathrm{FSR}=\frac{\lambda_{m}\lambda_{m+1}}{2d\sqrt{n_{cl}^{2}-n_{air}^{2}}},$$

The light transmission mechanism of the HCF can be explained by the anti-resonance principle. The transmission spectrum of the air layer confined in the middle of the HCF is extremely sensitive to changes in external factors, such as the change of in the air pressure inside the cavity that affect the refractive index (RI) of the air.
(3)$$S_{P}=\frac{d\lambda_{m}}{dP}=\frac{2d_{n_{cl}}}{m\sqrt{n_{cl}^{2}-n_{air}^{2}}}\frac{\partial n_{cl}}{\partial P}-\frac{2d_{n_{air}}}{m\sqrt{n_{cl}^{2}-n_{air}^{2}}}\frac{\partial n_{air}}{\partial P}+\frac{2\sqrt{n_{cl}^{2}-n_{air}^{2}}}{m}\frac{\partial d}{\partial P}$$
where *P* is the applied air pressure, and the influence of air pressure change on the cavity length of the optical fiber sensor can be ignored [30]. This is because the HCF is filled with air, the effective Young’s modulus is very small, and the longitudinal strain is almost determined by the outer cladding of silica. The refractive index changes of the HCF and the deformation of the tube wall can also be neglected [20].

Therefore, the drift of the resonance peak is mainly dominated by the change in the RI of the air. The sensitivity *S_P_* can be simplified into the following equation:(4)$$S_{P}=\frac{d\lambda_{m}}{dP}=-\frac{2d_{n_{air}}}{m\sqrt{n_{cl}^{2}-n_{air}^{2}}}\frac{\partial n_{air}}{\partial P},$$

As the pressure increases, the RI of air in the cavity also increases according to the following trend [4].
(5)$$n_{air}=1+\frac{2.8793\times 10^{-9}}{1+0.003671\times T}P$$

Therefore, if the temperature is kept at 25 °C, the pressure sensitivity can be calculated to be about 3.398 nm/MPa at 1500 nm in theory. This value is in agreement with the experimental results (−3.548 nm/MPa, −3.786 nm/MPa), which indicates that the RI of the air played a major role in the observed pressure sensitivity.

## 4. Experimental Results and Analyses

### 4.1. Experimental Setup

The experimental setup for temperature and air pressure sensing is shown in Figure 5. The light is output from a Broadband Light Source (BBS, FL-ASE-EB-D-2-FC/2-APC), a spontaneous radiation source with a spectral range of 1250~1650 nm. After the light passes through the sensor, it is finally connected to an Optical Spectrum Analyzer (OSA, AQ6370D), with an accuracy of ±0.02 nm. The interferometer is connected to the BBS and OSA at both ends, while the device is in the straightened state. All measurements were repeated three times.

### 4.2. Air Pressure Experiment of Single-Hole Sensor

Figure 6a shows an initial spectrum of the single-hole gas pressure sensor, in which the length and the inner diameter of the spliced HCF is 6 mm and 30 μm, respectively. It can be seen that the resonance peak intensity of the interference spectrum gradually rises with the increase in wavelength. As a result, the resonance peak near 1550 nm (Dip1), with an interference intensity of about 12 dB, is selected as the object of study.

As illustrated in Figure 5a, the single-hole fiber optic sensing head is placed inside a chamber with four ports. Its two ports are individually connected to BBS and OSA, while the remaining two ports are connected to an air pump and a barometer, respectively. In order to keep the chamber airtight, the four ports were sealed with epoxy AB adhesive. By pressing the air pressure pump, the air pressure inside the airtight chamber can be increased from 0.1 MPa to 0.8 MPa, while the Dip1 wavelengths are recorded every 0.1 MPa after the spectrum is stabilized. The spectral drift with air pressure is presented in the inset at the lower left corner of Figure 6b. When the air pressure inside the airtight chamber decreases from 0.8 MPa to 0.1 MPa, the Dip1 wavelengths are also recorded every 0.1 MPa. As shown in Figure 6b, the linear fits of the response curves almost overlap when the air pressure rises and falls, indicating the good repeatability. When the air pressure is rising, the sensitivity of the Dip1 wavelength is −3.548 nm/MPa with a linearity of 99.45%; and when the air pressure is falling, the sensitivity of the Dip1 wavelength is −3.429 nm/MPa with a linearity of 99.54%. It is worthy to indicate that the consistency of response curves has been verified by repeated measurements, in which the standard deviation of the multiple measurement results is too small to be distinguished by the OSA used in this work.

As shown in Figure 6a, four adjacent valleys in the wavelength range near 1500 nm to 1550 nm are selected for analysis. They are 1493.6 nm, 1515.4 nm, 1538 nm, and 1561 nm. Their adjacent intervals are 21.8 nm, 22.6 nm, and 23 nm. The FSRs are calculated from Equation (2) as 22.7 nm, 23.36 nm, and 24 nm, and it can be seen that the experimental and theoretical results are very close to each other.

### 4.3. Air Pressure Experiment of Ten-Hole Sensor

The transmission spectra of the ten-hole gas pressure sensor before and after femtosecond laser punching are shown in Figure 7a. The length of the HCF fused to the air sensor is 6.2 mm, and its inner diameter is still 30 μm. The perforated device was annealed at 600 °C to relieve the stress induced by laser ablation. The intensity of the transmission spectrum does not change obviously before and after annealing, while the wavelength of the resonant peak drifted significantly.

The sensitivity measurement of the ten-hole gas pressure sensor is almost the same as that of the single hole sensor. At first, the air pressure in the airtight chamber slowly rises from 0 MPa to 0.6 MPa, while the Dip2 wavelengths (see Figure 7a) are recorded every 0.1 MPa. The spectral drift with air pressure is presented in the inset at the lower left corner of Figure 7b. Then, the air pressure slowly decreases from 0.6 MPa to 0 MPa. At the same time, the Dip2 wavelengths are also recorded every 0.1 MPa. Figure 7b shows linear fits of the response curves. The linearities of the rising and falling curves are 99.43% and 99.47%, and the sensitivities of the Dip2 wavelength are −3.643 nm/MPa and −3.786 nm/MPa, respectively.

### 4.4. Comparison Experiments of Response Times of Sensors

Response time is an essential parameter to judge the performance of the fiber optic air pressure sensor. Here, the response time of one-hole and ten-hole air pressure sensors are measured with the experimental setup in Figure 5a. Firstly, the sensing head was placed in the airtight chamber, and the Dip1 wavelength was recorded when the air pressure was set at 0 MPa. Then, the chamber pressure was rapidly increased to 0.3 MPa, and the time of Dip1 drift after the pressure reach 0.3 MPa was recorded. After holding for 30 s, the chamber pressure was rapidly dropped from 0.3 MPa to 0 MPa, and the time of Dip1 drift after the pressure reach 0 MPa was recorded again. The above measurements were repeated six times, and the ramp-up and the ramp-down average response times are illustrated in Figure 8. It can be seen that the response speed of the ten-hole sensor is twice as fast as that of the single-hole sensor. We expect that the faster response speed is caused by the smaller resistance of air entering into the HCF with more holes. It is worthy to note that the response time would be shorter by machining an open microcavity directly on the fiber crossing the core layer, but this microcavity will seriously reduce the robustness of fiber sensor. Multiple micro-holes in an HCF offer a pathway to raise the response speed of air pressure sensors but keep its robustness to high pressure environment.

### 4.5. Temperature Experiment

Figure 5b shows a diagram of the temperature experimental setup of the single-hole sensor, its sensing head was placed in a temperature-controllable tubular heating furnace, and the temperature was gradually increased from 30 °C to 130 °C by the program setting, while the spectral data were recorded every 10 °C, as displayed in the upper left inset of Figure 9. When the temperature rose, the wavelength of the resonance valley Dip1 of the single-hole sensor was red-shifted, and its intensity barely changed. The linear fit of the resonance valley Dip1 is 98.37%, its temperature sensitivity is 10.3 pm/°C, with the cross-sensitivity of temperature being 2.9 kPa/°C (0.0103/3.548 = 0.0029 MPa/°C), indicating that the device has a very small cross-sensitivity to temperature changes.

## 5. Conclusions

Single-hole and ten-hole HCF gas pressure sensors were demonstrated by femtosecond laser drilling in the HCFs sandwiched between two SMFs. The air pressure sensing experiments were performed respectively, showing that the air pressure sensitivity of the single-hole sensor reached the maximum at −3.548 nm/MPa with a linear fit of 99.45%, and the air pressure sensitivity of the ten-hole sensor achieved the maximum at −3.786 nm/MPa with a linear fit of 99.47%. These experimental results were both close to the theoretical values. The air sensitivities of the single-hole and ten-hole gas pressure sensors are similar, but the response speed of the ten-hole fiber-optic sensor is approximately twice as fast as that of the single-hole one. In addition, the temperature sensitivity of the single-hole air sensor is only 10.3 pm/°C, and the cross-sensitivity of temperature and air pressure is 2.9 kPa/°C, indicating that the effect of temperature fluctuations on the sensor is negligible compared to its air pressure sensitivity. Due to the advantages of low cost, easy fabrication, high air pressure sensitivity but insensitive to temperature, small size, fast response time, and good structural robustness, the device with multi-holes can offer a wide range of applications in high-pressure measurements in extreme environments.

## Figures and Tables

**Figure 1 micromachines-14-00101-f001:**
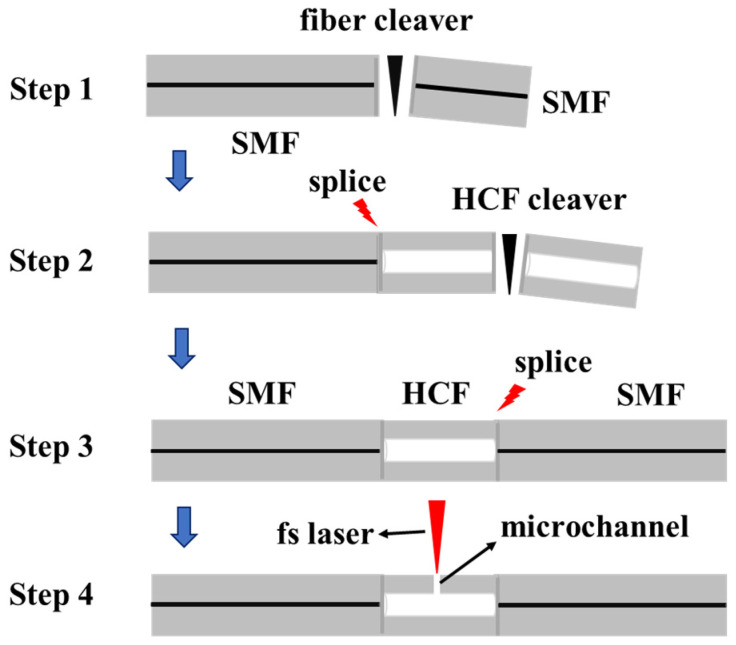
Schematic diagram of a fusion splicing process between SMF and HCF.

**Figure 2 micromachines-14-00101-f002:**
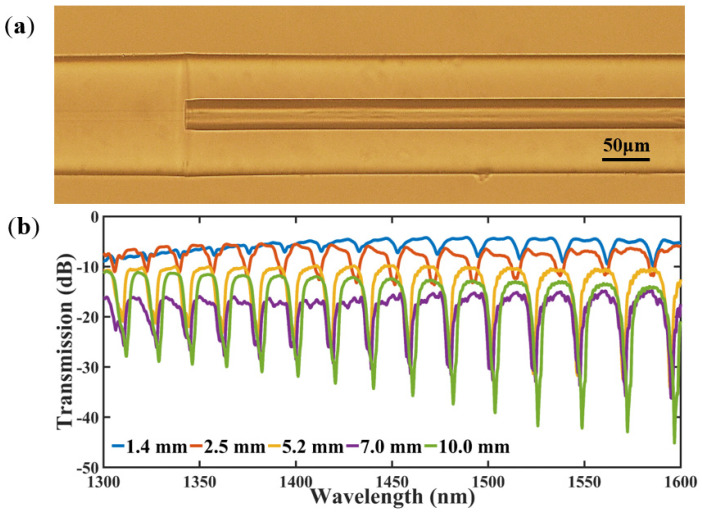
(**a**) A microscope image of a junction between a SMF and an HCF, (**b**) Interference spectrum corresponding to different lengths of HCF.

**Figure 3 micromachines-14-00101-f003:**
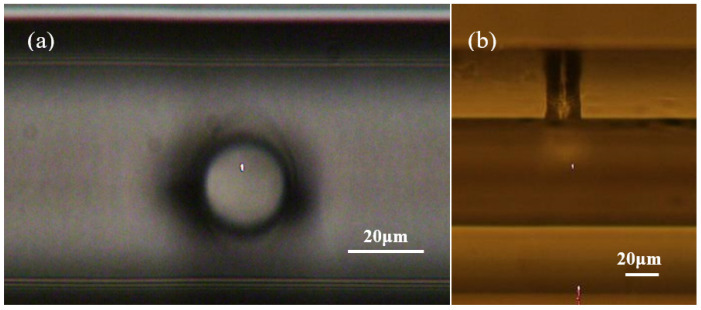
Microscopic images of single hole drilled in a HCF. (**a**) Front view; (**b**) Side view.

**Figure 4 micromachines-14-00101-f004:**
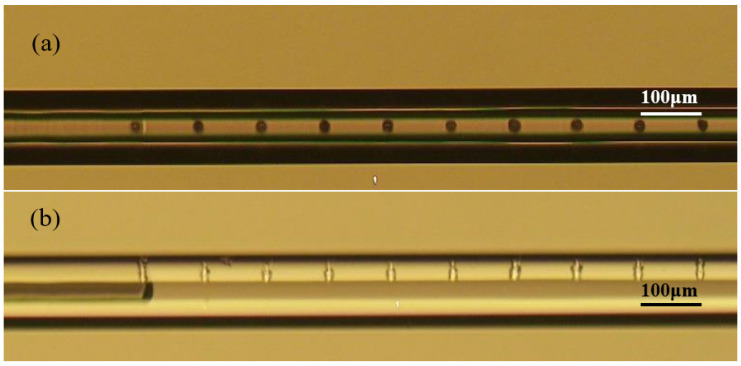
Microscopic images of ten micro-holes drilled in an HCF. (**a**) Front view; (**b**) Side view.

**Figure 5 micromachines-14-00101-f005:**
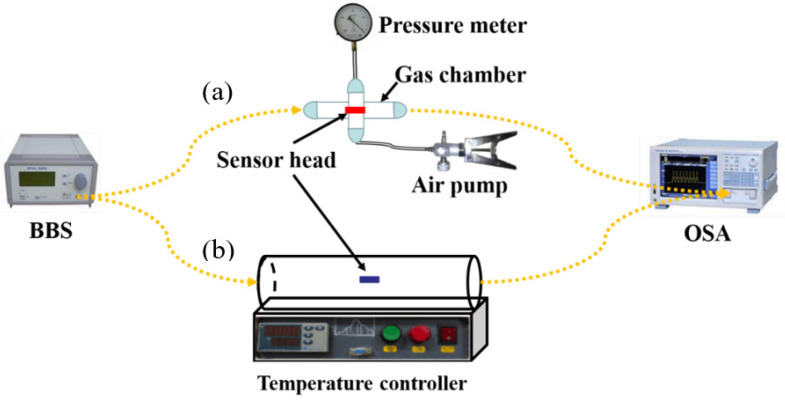
Schematic diagram of the experimental setup. (**a**) Air pressure sensing unit; (**b**) Temperature sensing unit.

**Figure 6 micromachines-14-00101-f006:**
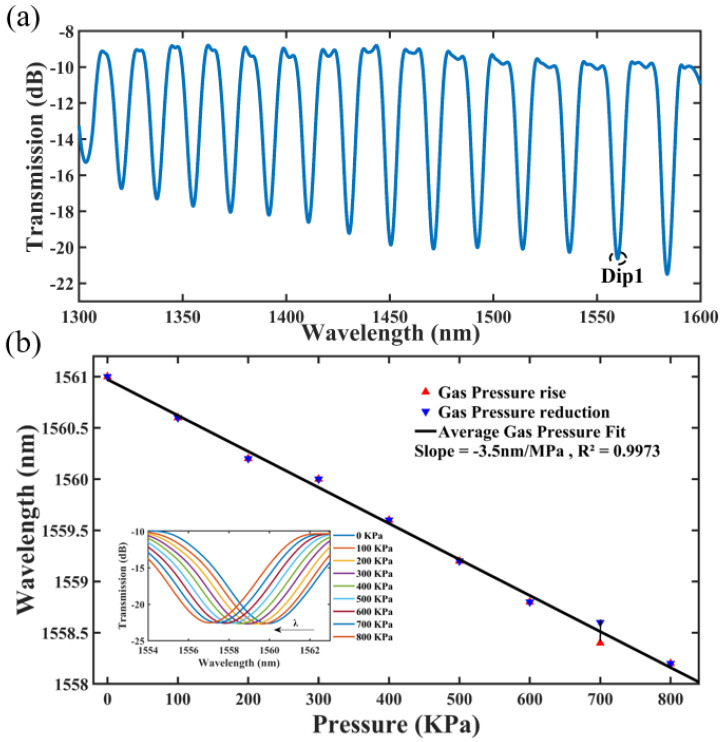
Single-hole sensor for air pressure. (**a**) Initial spectrum; (**b**) Spectral drift and linear fit.

**Figure 7 micromachines-14-00101-f007:**
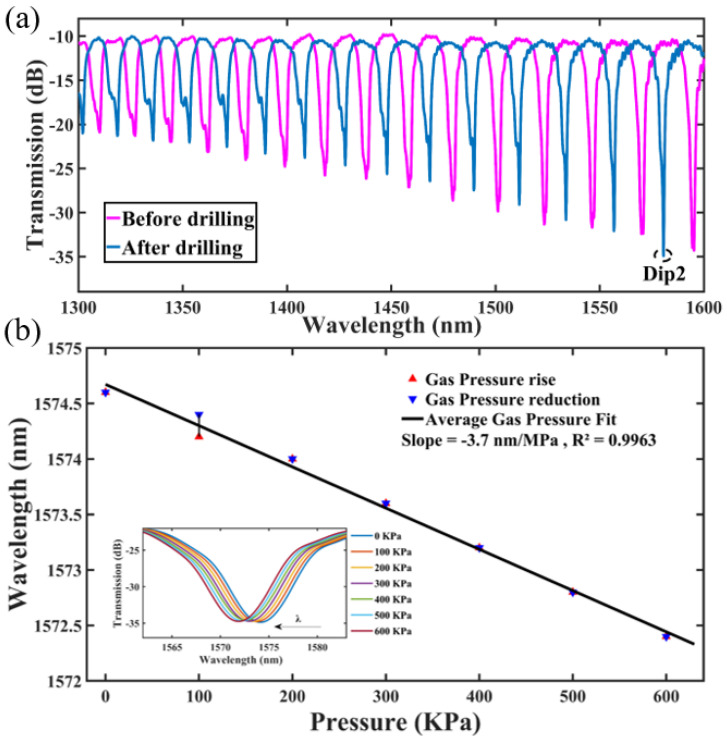
Air pressure sensing experiment data of ten-hole sensor. (**a**) Initial spectrum; (**b**) Spectral drift and linear fit.

**Figure 8 micromachines-14-00101-f008:**
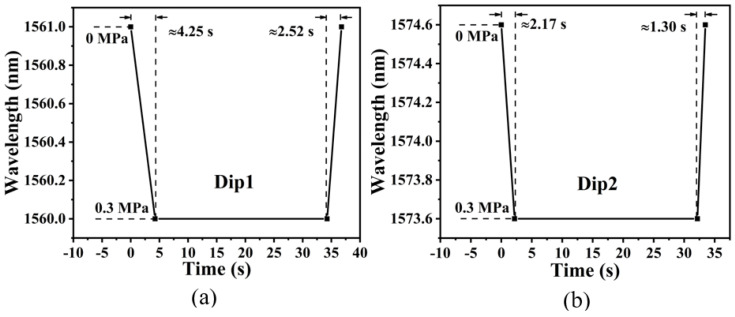
Comparison graphs of response times: (**a**) Response time of single-hole sensor; (**b**) Response time of ten-hole sensor.

**Figure 9 micromachines-14-00101-f009:**
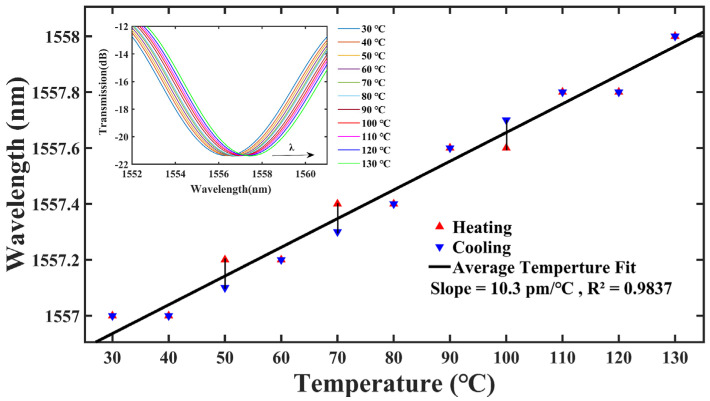
Spectral drift and linear fit of temperature sensitivity.

## Data Availability

Data underlying the results presented in this paper are not publicly available at this time but may be obtained from the authors upon reasonable request.

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
