# Peer review of "Fabricating Air Pressure Sensors in Hollow-Core Fiber Using Femtosecond Laser Pulse"

_micromachines, 2022, doi:10.3390/mi14010101_

Round 1

Reviewer 1 Report

Comments for the paper to improve:

1. Paper title about "response time", while the paper emphasized on sensitivity as well. Response is not impressively rapid. Suggest to modify the title.

2. Can authors tell more about the fiber cutting knife? Is laser cutting better? Any other possible fabrication method to get holes on fiber?

3. From the side view, the holes look quite rough. Is the hole working only as air path, or effecting the optic wave path as well? I think it could lead to a loss of optic wave transmission which reduced the spectra resonance quality factor.

4. What's the quality factor of the spectra? Comparing to the sensitivity, I don't think it's high enough. And what’s the resolution of wavelength detection?

5. Can authors add in the error bars in Fig. 6(b)? Two “Dip 1” in the plot. How about the response in the rang of lower than 1 atm?

6.Can the experimental data match well with the theoretical resonant wavelength (Eq.1) and sensitivity (Eq.3)? It's better to have a comparison between theory and experiment, otherwise, the theory part doesn't make sense to be shown here.

7.Fig. 9: error bar suggested. How does the air pressure sensitivity change in different temperature? 

8 How the number of holes and hole diameter effect the sensitivity, response time, et al?

Reviewer 2 Report

In this manuscript, the authors present the study of hollow optical fiber sensors with different hole numbers. This manuscript's novelty and scientific impact qualify for publication in the Micromachines (MDPI). Nevertheless, before publishing, I still have the following suggestions for the authors’ reference.

1.     The title “...with femtosecond laser perforation” seems inappropriate. A femtosecond laser is a tool for punching the optical fiber to fabricate a hollow fiber sensor, but not a characteristic of the sensor.    

2.     In Figure 1, HACF is a typo.

3.     In Figures 6 & 7, the authors do not state the number of experiments and do not indicate error bars. How is the reproducibility?

4.     The authors highlight that they are the first to compare the response times of single-hole and ten-hole optical fiber sensors. If so, I suggest the authors achieve a complete comparison, such as 1-hole, 5-hole, 10-hole, and 15-hole sensors, to obtain a generalized conclusion.  

Reviewer 3 Report

My general evaluation for the article titled “Study on response time of fiber optic air pressure sensor with femtosecond laser perforation” is as follows.

  It is a good study in the field of “Response time of fiber optic air pressure sensor”. It is seen that the study was organized and written in accordance with its purpose. This study can be published in your journal as it is, but making the following corrections will strengthen the article.

1. The abstract should be strong. The abstract should be revised. The technique-method can be stated briefly. However, only results are given in the summary. The purpose is not clearly stated.

2. In general, the English language of this article should be corrected. Professional help is recommended.

3. I think the Introduction section is good.

4. Which Microscopic device the image was taken in Figure 3-4?. It can be specified. (model of devices)

5. Figure 1 in the "Fabrication of sensors" section should be given more clearly. (For example, fabrication stages can be specified.)

6. Conclusions section should be developed. The superiority-difference between this article from other existing studies should be clearly stated.

Reviewer 4 Report

The manuscript reports the study on the response time of a perforated fiber optic air pressure sensor. The perforation was done by the femtosecond laser method, making the device able to be fabricated at a reasonable cost. The performance of fiber optic air pressure sensors with different perforation numbers was compared, and the experimental results show that the device with more perforation numbers has a faster response speed, while the sensitivity remains almost the same and temperature-dependency is negligible. The experimental work was rigorously done.

Although the topic falls within the journal's scope, the concept of femtosecond laser perforated fiber optic air pressure sensor was already reported a few years ago. The increased perforation number did increase the response speed compared with the single perforation device, however, the response speed is still in the range of a few seconds, while device designs fabricated with other cost-efficient methods are in the microsecond range (example: DOI 10.1109/JSEN.2021.3068456). The significance of this work is not revealed, therefore, the referee does not suggest this manuscript be published in the journal Micromachines in its current form, unless the authors can highlight the importance of their work in a revised version.

In addition to the comments above, some minor remarks:

1. The authors mentioned in the conclusion that ‘the fiber optic air pressure sensors may find broad applications in the fields of weather monitoring as well as high pressure measurement’. As mentioned above, it might help the authors justify the importance and potential of their work by clarifying the requirements from the mentioned application fields.

2. The micro-hole shown in Fig. 3 looks sharp, while most of the micro-holes in Fig. 4 look rather coarse, especially holes No. 3/7/9/10. Please explain the cause of the low-quality micro-holes, and the potential influence on the device performance.

3. In Fig. 4(b), there is some defect-like structure under micro-hole No.1, please add an explanation for this defect.

4. Although P normally stands for pressure in equations, it’s still required to clearly define it to avoid any misinterpretation (please add a definition close to Eq. 2).

5. There are a few grammar errors, language needs to be checked.

Round 2

Reviewer 1 Report

Thanks much for the authors’s response and additional work! I have some follow-up questions or comments on the authors’ responses to the first-round review questions:

Q2: What’s the accuracy of using a fiber cleaver to drill the holes w.r.t the dimension and distance? The question is related to the accuracy of the resonance in the spectra due to the sensor fab.

Q3: It will be interesting to see how the quality factor can be improved by using more accurate fabrication manners like wet/dry etching. If this can be much improved, it could be a good technology for industrial use.

Q4: when you say “is much more than the minimum resolution”, what do you expect about the pressure resolution you want to achieve, or needed? -what’s the targeting situation you were planning for the application with this sensor?

Q5: The error-bar I mean is not the two-data variation of rise and fall, but repeated groups of measurement. Meanwhile, your experimental setup should give air pressure with much smaller intervals, so you should be able to obtain seas of data which such a big pressure range, while you only have an internal of 100 kPa. From the 0.02nm resolution and 3.786nm/MPa sensitivity you mentioned, you’d better have data with 5kPa or smaller interval to confirm the sensor performance. This is the most important figure of this papers, however, is not persuasive at the current version.

Q6: I suggest to put the comparisons in the experiment session near Fig.6, rather than in the theory one. Btw, looks like the difference is about 1nm between theory and experiment, this means ~300 kPa difference, which is not “”close” at all. Could you please comment more and explain it?

Q7: I like the temperature-dependent study, and that’s great the sensor is not quite sensitive to temperature variation. While, I still suggest you do the same thing as I suggest in Q5, which is more temperature data points, and more repeated measurement.

Q8: This would be a great study if you could explore more on that, the wet etching process could be able to help you vary lots of situations.

Reviewer 2 Report

Even though the authors have revised the manuscript, the degree of novelty in this study is not high enough to publish in the Micromachines. There have been many similar papers published. In particular, this manuscript's illustration is very similar to the paper published in Optics Express (Vol. 27, No. 16, pp. 22181-22189, 2019). I don't see any noticeable difference. Although this manuscript presents a ten-hole sensor for comparison, the authors didn’t explain why the response speed of the ten-hole sensor is twice as fast as that of the single-hole one.

Besides, there are some typos and unlikely illustrations in the manuscript:

1.     Line 152: …repeated three times.

2.     Line 230: Single-hole and ten-hole HCF…

In conclusion, the authors say “…the device with multi-holes by femtosecond laser drilling can offer a wide range of applications in different fields such as weather monitoring and high-pressure measurements in extreme environments.” The femtosecond laser is a tool for punching the optical fiber to fabricate a hollow fiber. Unless the author mentions the particularity of femtosecond lasers in this study, I don't see why it should be highlighted here. Besides, I don’t understand why the multi-hole fiber sensor is better for weather monitoring and high-presure measurements, but not the single-hole one.

Reviewer 4 Report

All concerns have been addressed.

Author Response

Thank you very much for your positive comments.

Round 3

Reviewer 2 Report

The authors’ responses still can not convince me to believe that this study has any noticeable difference from Gao’s paper (Optics Express, Vol. 27, No. 16, pp. 22181-22189, 2019). I still maintain the previous decision and do not recommend this manuscript for publication.